# Chromatin sequesters pioneer transcription factor Sox2 from exerting force on DNA

Tuan Nguyen[1,2], Sai Li[1], Jeremy T-H Chang[1,2], John W. Watters [1], Htet Ng[1], Adewola Osunsade[3], Yael David [3] & Shixin Liu [1✉]

Biomolecular condensation constitutes an emerging mechanism for transcriptional regulation. Recent studies suggest that the co-condensation between transcription factors (TFs) and DNA can generate mechanical forces driving genome rearrangements. However, the reported forces generated by protein-DNA co-condensation are typically below one piconewton (pN), questioning its physiological significance. Moreover, the force-generating capacity of these condensates in the chromatin context remains unknown. Here, we show that Sox2, a nucleosome-binding pioneer TF, forms co-condensates with DNA and generates forces up to 7 pN, exerting considerable mechanical tension on DNA strands. We find that the disordered domains of Sox2 are required for maximum force generation but not for condensate formation. Furthermore, we show that nucleosomes dramatically attenuate the mechanical stress exerted by Sox2 by sequestering it from coalescing on bare DNA. Our findings reveal that TF-mediated DNA condensation can exert significant mechanical stress on the genome which can nonetheless be attenuated by the chromatin architecture.

[1] Laboratory of Nanoscale Biophysics and Biochemistry, The Rockefeller University, New York, NY, USA. [2] Weill Cornell/Rockefeller/Sloan Kettering Tri-Institutional MD-PhD Program, New York, NY, USA. [3] Chemical Biology Program, Sloan Kettering Institute, Memorial Sloan Kettering Cancer Center, New York, NY, USA. ✉email: shixinliu@rockefeller.edu

Transcription factors (TFs) bind specific DNA sequences within the genome to regulate the activity of the transcription machinery[1]. In recent years, a new paradigm for transcriptional control has emerged in which the intrinsically disordered regions (IDRs) of certain nuclear proteins drive the formation of biomolecular condensates and phase-separated sub-compartments[2–5]. These nuclear compartments, or transcriptional hubs, connect enhancers to promoters, recruit the RNA polymerase and its regulators, and control gene expression in a dynamic fashion[6,7]. Notably, some TFs have been shown to form co-condensates with DNA[8–11], which ensnare a certain amount of DNA in the condensed phase and thus exert tension on the outside free DNA[12]. The forces generated by this mechanism were reported to be in the sub-piconewton (pN) range[8,9,13]. Whether the mechanical effect driven by TF:DNA co-condensation is strong enough to be relevant in the nuclear milieu remains to be answered. Moreover, the genomic DNA in eukaryotic nuclei is spooled by histone proteins to form nucleosomes and further organized into higher-order chromatin structures[14]. How the chromatin organization impacts TF condensation and its force-generating capacity is still unclear.

In this report, we employed single-molecule imaging and manipulation to compare the mechanical effects of TF:DNA co-condensation on bare DNA versus nucleosomal DNA. We chose Sox2 as the model TF, which belongs to the high mobility group (HMG) superfamily of proteins that bind DNA and nucleosomes to induce structural changes in chromatin and cell fate transitions[15]. We observed the real-time formation of Sox2:DNA co-condensates that exert surprisingly high intra-strand and inter-strand mechanical stress. The maximum condensation force that Sox2 can generate was measured to be ~7 pN, an order of magnitude higher than those previously reported for other DNA-binding proteins[8,9,13]. Remarkably, when nucleosomes are present on DNA, the mechanical effects of Sox2:DNA condensation are drastically reduced. These results suggest that nucleosomes function more than just DNA packaging units, but also as mechanical sinks to regulate the force generated by protein:DNA co-condensates.

## Results

**Sox2 forms co-condensates with DNA.** We used the bacteriophage λ genomic DNA (λDNA) as a model DNA substrate for this study. Individual λDNA molecules were immobilized on a glass surface via biotin-streptavidin linkage, stained with the YOPRO1 dye that intercalates into the DNA backbone, and imaged with total-internal-reflection fluorescence microscopy (TIRFM) (Fig. 1a)[16]. Double-tethered λDNA molecules exhibited a distribution of end-to-end distances due to heterogeneous surface-anchoring of the two ends. Molecules with short end-to-end distances displayed larger transversal fluctuations—due to more slacks—than those with long end-to-end distances (Fig. 1a). After flowing in Cy5-labeled recombinant full-length human Sox2 (Supplementary Fig. 1), we observed the formation of Sox2 foci on the DNA (Fig. 1b), which contains numerous Sox2 binding motifs across its native sequence (Supplementary Fig. 2). These foci displayed mobility on the DNA as well as fusion and splitting events (Supplementary Fig. 3a), indicating liquid-like properties[2,17]. Upon Sox2 binding and foci formation, we also observed that the fluorescence signal of the DNA transitioned from a relatively uniform distribution to a few clusters that colocalized with the Sox2 foci (Fig. 1b). This was particularly apparent in the DNA strands with a short end-to-end distance.

Once nucleated, the Sox2 foci on DNA were long-lived, and the fluorescence intensities of both Sox2 and DNA at the foci increased with time until reaching a steady state (Fig. 1c, d). Interestingly, we observed a loss of the fluctuating motion in the

DNA concurrent with Sox2 foci formation (Fig. 1a, b). Indeed, the average DNA envelope width—a measure for the degree of transversal fluctuations—was significantly reduced in the presence of Sox2 (Fig. 1e, f). Even though the DNA envelope is wider for molecules with shorter end-to-end distances in the absence of Sox2 as expected, the addition of Sox2 reduced the envelope width for all double-tethered molecules to the same level (Fig. 1e). These findings can be rationalized by an ability of Sox2 to form co-condensates with DNA. As more DNA being pulled into the condensates, the previously slacked DNA transitioned into a tensed state.

In addition, we observed a fraction of λDNA molecules that were tethered to the surface at only one end (Fig. 1g), likely because the other biotinylated end did not find a streptavidin to bind during flow injection. Without Sox2, these single-tethered DNA molecules displayed random fluctuations characterized by a measurable radius (Fig. 1g, i). The addition of Sox2 again visibly suppressed such fluctuations (Supplementary Movie 1)—most likely due to co-condensation with DNA—resulting in a significantly decreased average fluctuation radius (Fig. 1h, i). Altogether, these results demonstrate that Sox2 and DNA form co-condensates wherein proteins and DNA accumulate, reducing the amount of free DNA outside the condensates.

**Sox2:DNA co-condensation exerts mechanical stress on DNA.** The loss of fluctuations in both single- and double-tethered DNA suggests that Sox2-induced condensation generates mechanical tension within the DNA. These effects were recapitulated using unlabeled wild-type Sox2, ruling out the possibility of labeling artifact (Supplementary Fig. 4a, b). In accordance with this notion but nonetheless unexpectedly, we observed that a significant population of double-tethered DNA underwent sudden breakage after losing slacks (Fig. 2a, b, Supplementary Fig. 4c, and Supplementary Movie 2). The breakage was accompanied by a rapid collapse of the Sox2 and DNA fluorescence signals into the two tethered ends (Supplementary Movie 2). Notably, these breakage events occurred over a time window that coincided with the formation of Sox2:DNA co-condensates and became much less frequent as the mobility of Sox2 foci decreased over time (Supplementary Fig. 3b–d). In contrast, virtually no DNA breakage was observed in the absence of Sox2 (Fig. 2b), or for single-tethered λDNA, where the tension can be released from the free end, even after the addition of Sox2 (Supplementary Fig. 5a).

We then explored other factors besides condensation-induced tension that could contribute to the tether breakage. We found that the fraction of broken DNA tethers was not significantly affected by the duration of laser exposure in our experiments (Supplementary Fig. 5b). Given the known effect of DNA intercalating dyes on the mechanical properties of DNA[18,19], we washed out YOPRO1 prior to the addition of Sox2 and observed a lower fraction of ruptured DNA (Supplementary Fig. 5c). To evaluate whether nicks that inevitably exist in these λDNA samples played a role in tether breakage, we treated the DNA with T4 ligase and observed fewer breakage events upon Sox2 condensate formation (Supplementary Fig. 5d). Based on these results, we speculate that the DNA breakage observed in the TIRFM experiments resulted from a combination of tension generated by Sox2:DNA co-condensation and mechanical instability in the DNA substrate due to nicks and the intercalating dye. Nonetheless, the breakage fraction is still a useful proxy for the magnitude of mechanical tension to compare different proteins and substrates if the same imaging conditions and DNA batch are used.

To examine whether other DNA-binding proteins can exert the same level of tension on DNA, we repeated the above TIRFM

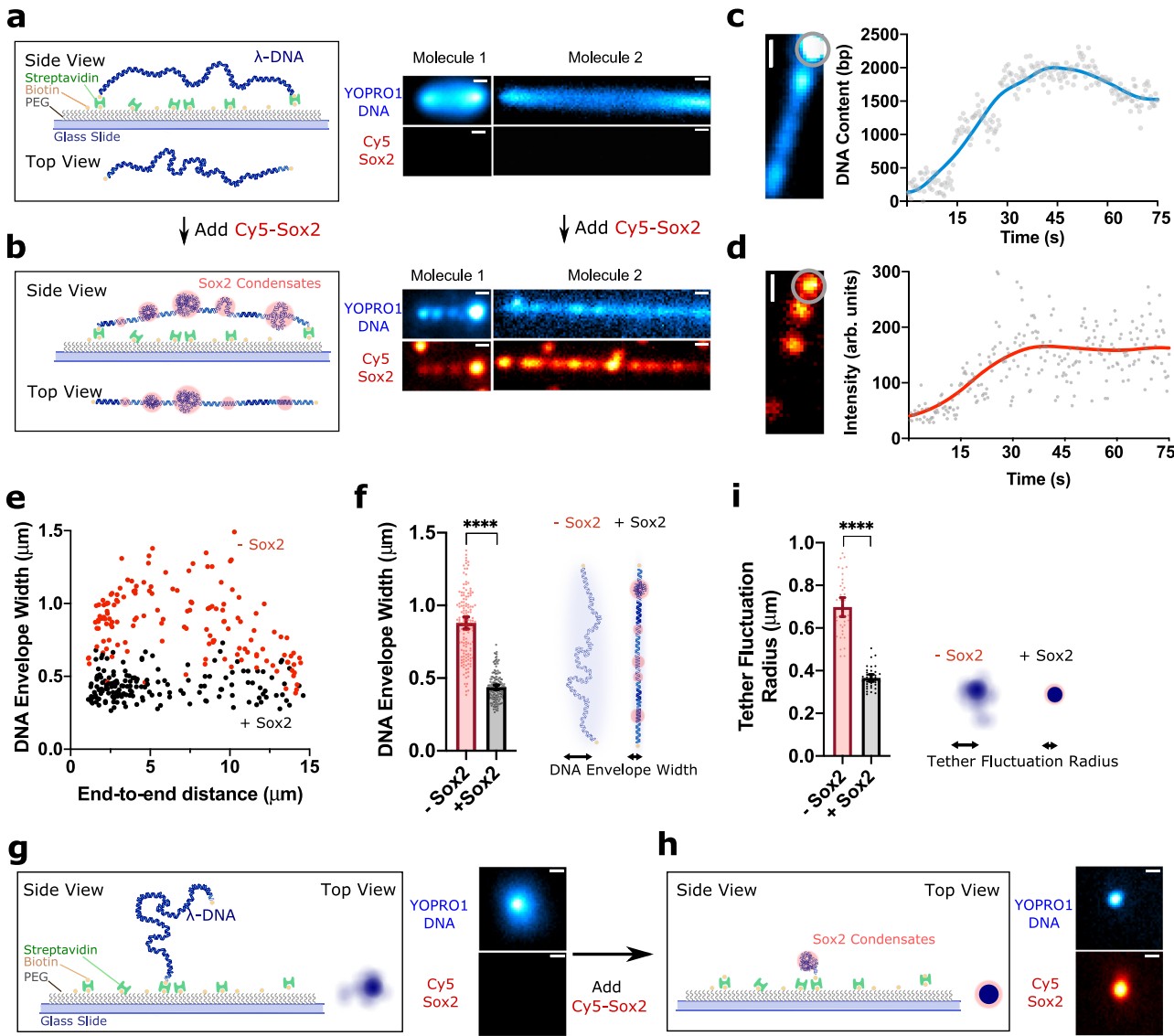

**Fig. 1 Sox2 forms condensates on DNA. a** (Left) Schematic of a double-tethered λDNA. (Right) Two example images of double-tethered λDNA molecules with different end-to-end distances (among 4 independent experiments). DNA was stained with YOPRO1 (20 nM) and imaged by TIRFM. Shown are time-averaged projections over a 75-s period. Scale bar, 0.5 μm. **b** Schematic and time-averaged projections of the same two λDNA molecules as in **a** when incubated with 10 nM Cy5-labeled Sox2. Scale bar, 0.5 μm. **c** Real-time tracking of the DNA content (YOPRO1 fluorescence intensity converted to the amount of DNA base pairs) within a condensate (circled region) on a double-tethered λDNA (among 4 independent experiments). Scale bar, 0.5 μm. **d** Corresponding changes in the Sox2 intensity within the same circled region as in **c**. Scale bar, 0.5 μm. **e** DNA envelope width as a function of the end-to-end distance of double-tethered DNA measured in the absence ($n = 147$) or presence of Sox2 ($n = 162$), where $n$ represents the number of DNA molecules analyzed. **f** Bar graph and cartoon showing a reduction in the mean DNA envelope width averaged over all the molecules shown in **e** upon Sox2-mediated co-condensation. Error bars denote 95% CI. Significance was obtained using an unpaired two-sample $t$ test (****$P < 0.0001$). **g** Schematic and time-averaged projection of a single-tethered λDNA stained with YOPRO1 displaying random fluctuations. Scale bar, 0.5 μm. **h** Schematic and time-averaged projection of the same single-tethered λDNA as in **g** showing Sox2-mediated condensation. Scale bar, 0.5 μm. **i** Bar graph and cartoon showing the mean fluctuation radius of single-tethered DNA molecules in the absence ($n = 38$) or presence of Sox2 ($n = 37$). Error bars denote 95% CI. Significance was obtained using an unpaired two-sample $t$ test (****$P < 0.0001$). Source data are provided as Source Data Fig. 1.

assay with another abundant nuclear protein, the human linker histone H1.4 (referred to as H1 hereafter). H1 is known to form co-condensates with DNA[10,13,20]. We found that H1:DNA co-condensation also reduced the double-tethered DNA envelope width and single-tethered DNA fluctuation radius (Supplementary Fig. 6c–e). However, H1-mediated DNA condensation resulted in much fewer DNA breakage events compared to Sox2-mediated condensation (Fig. 2b), suggesting that H1 generates a lower force on DNA.

We then sought to examine whether Sox2:DNA co-condensation can generate inter-strand tension. In the absence of Sox2, the neighboring λDNA strands immobilized in proximity of each other fluctuated independently. Upon the addition of Sox2, these strands lost slack and joined one another through the fusion of Sox2 foci (Fig. 2c and Supplementary Movie 3). In some cases, we observed successive severing and joining of DNA located nearby (Fig. 2d and Supplementary Movie 4). Together, these results suggest that Sox2 condensates exert force on DNA

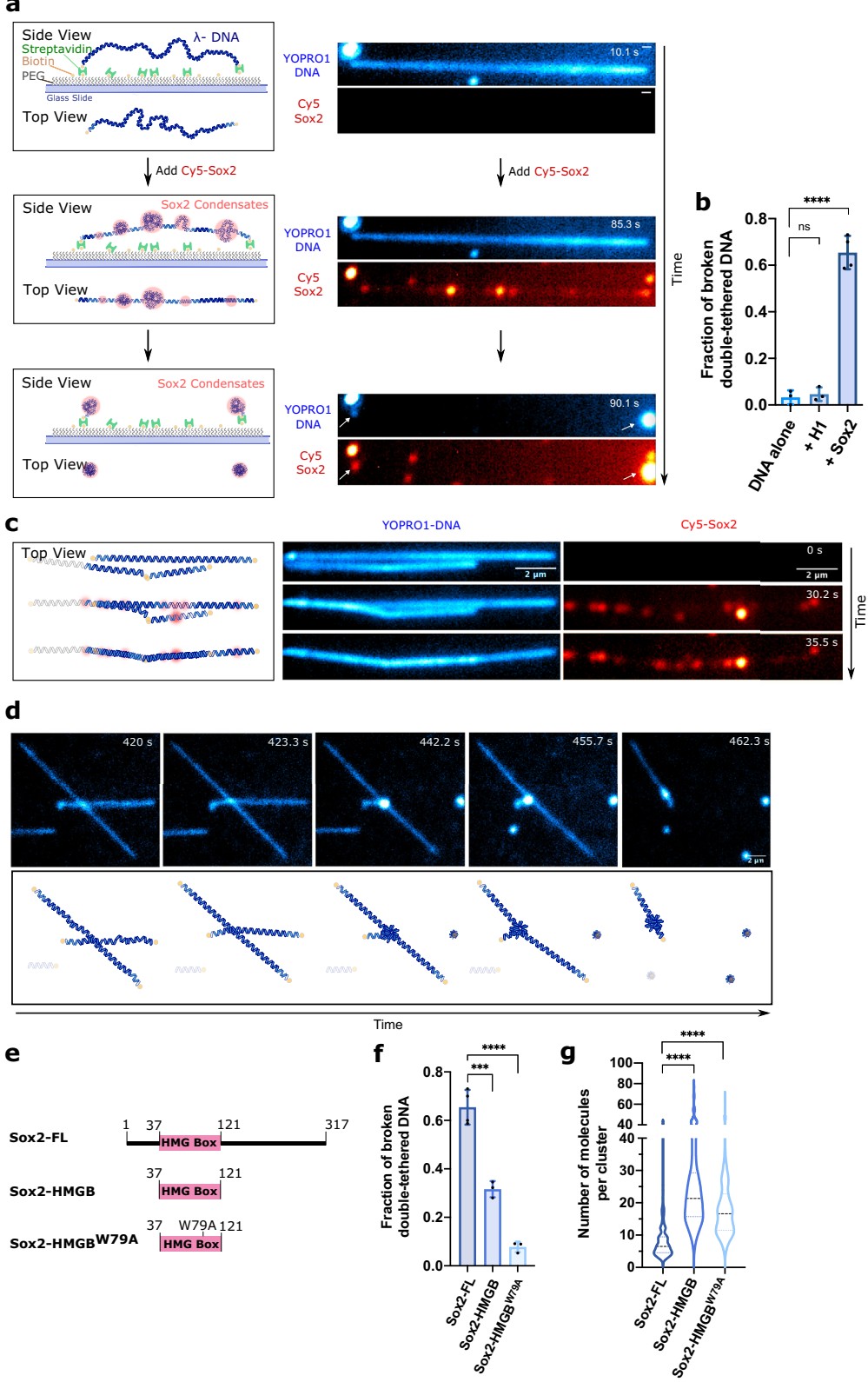

both within the same strand (when both ends are anchored) and between multiple strands.

**IDRs of Sox2 are dispensable for condensate formation but required for force exertion.** Sox2 contains N- and C-terminal IDRs flanking the globular DNA-binding HMGB domain[21]. To gain insight into the molecular mechanism that underlies the

capacity of Sox2 to form co-condensates with DNA, we generated and fluorescently labeled Sox2 truncations (Fig. 2e and Supplementary Fig. 1). We first examined a Sox2 construct that contains only its HMGB domain without the IDRs (Sox2-HMGB). Somewhat unexpectedly, similar to the full-length Sox2 (Sox2-FL), Sox2-HMGB also formed foci on λDNA strands—both doubly and singly tethered—along with a concomitant loss of DNA

**Fig. 2 Sox2:DNA co-condensation exerts intra- and inter-strand mechanical stress. a** Schematic (left) and time-lapse snapshots (right) showing Sox2 condensate formation on a double-tethered λDNA and the subsequent breakage event upon which both DNA and Sox2 signals collapsed to the two tethered ends (white arrows) (among 4 independent experiments). Scale bar, 0.5 μm. **b** Fraction of double-tethered λDNA molecules that broke after 15 min without any protein ($n = 251$), with 10 nM H1 ($n = 150$), or with 10 nM Sox2 ($n = 379$). Error bars denote standard deviation. Data are collected from at least three fields of view. Significance was obtained using a one-way ANOVA with Dunnett's test for multiple comparisons (ns, $P = 0.9267$; ****$P < 0.0001$). **c** Schematic (left) and time-lapse snapshots (right) showing multiple adjacent DNA strands (among 4 independent experiments) being joined upon Sox2 condensate formation. **d** Time-lapse snapshots (top) and cartoon illustrations (bottom) showing a series of DNA breaking and joining events occurring among multiple λDNA strands in the presence of Sox2. **e** Schematic of different Sox2 constructs used in this study. **f** Fraction of double-tethered λDNA molecules that broke after 15 min of incubation with Sox2-FL ($n = 379$), Sox2-HMGB ($n = 357$), or Sox2-HMGB[W79A] ($n = 297$). Error bars denote standard deviation. Significance was obtained using a one-way ANOVA with Dunnett's test for multiple comparisons (***$P < 0.001$; ****$P < 0.0001$). **g** Violin plot showing the distribution of the number of Sox2 molecules within each cluster for Sox2-FL ($n = 167$), Sox2-HMGB ($n = 168$), or Sox2-HMGB[W79A] ($n = 155$), where $n$ represents the number of clusters analyzed. Significance was obtained using a one-way ANOVA with Dunnett's test for multiple comparisons (****$P < 0.0001$). Source data are provided as Source Data Fig. 2.

fluctuations (Supplementary Fig. 7). This observation indicates that the IDRs of Sox2 are not required for its co-condensation with DNA. However, Sox2-HMGB took a much longer time to form the same amount of DNA condensation (measured through the loss of fluctuations) ($T_{condense} = 184 \pm 45$ s) compared to Sox2-FL ($T_{condense} = 30 \pm 4$ s). Sox2-HMGB:DNA co-condensation also resulted in significantly fewer DNA breakage events (Fig. 2f). We next introduced a single-residue mutation W79A to Sox2-HMGB, generating Sox2-HMGB[W79A]. Consistent with previous results[22], Sox2-HMGB[W79A] displayed a diminished DNA-binding activity (Supplementary Fig. 8). Nevertheless, it still retained the ability to form co-condensates with DNA (Supplementary Fig. 7), albeit with even slower condensation kinetics ($T_{condense} = 251 \pm 59$ s). This point mutation further attenuates the condensation-dependent mechanical tension exerted on DNA (quantified by the fraction of broken double-tethered DNA) compared to both Sox2-FL and Sox2-HMGB (Fig. 2f). Notably, the mechanical effect of Sox2:DNA co-condensates is not directly correlated with their size, as both Sox2-HMGB and Sox2-HMGB[W79A] foci on average contained more Sox2 molecules—estimated from the brightness of the fluorescent foci—than Sox2-FL foci (Fig. 2g). Together, these results demonstrate that the HMGB domain alone can mediate Sox2:DNA co-condensation, but the high mechanical stress on DNA is largely driven by the IDRs of Sox2.

**Quantification of the force generated by Sox2:DNA co-condensation.** Next, we sought to quantitatively measure the force exerted by Sox2:DNA co-condensates on the DNA strand. Using optical tweezers combined with scanning confocal microscopy[23], we tethered a single λDNA molecule between two optically trapped beads, moved the tether in its relaxed form (i.e., zero applied force) to a channel containing Cy3-labeled Sox2, and monitored the force on DNA as a function of time. We first conducted experiments in a passive mode by keeping the trap positions fixed (Fig. 3a). As Sox2 foci appeared and accumulated on the DNA tether, the force reading concurrently increased. Both fluorescence and force values reached a plateau after 10–20 s (Fig. 3b). Force generation requires the presence of Sox2, and the force plateau level is dependent on the concentration of Sox2 in solution (Fig. 3c) with a maximum value of ~7 pN measured at the highest Sox2 concentration tested (500 nM).

We then conducted force-clamp experiments in which the tethered DNA was incubated with Sox2 at a constant force by adjusting the trap separation (Fig. 3d). We observed that, with a force clamp set at 0.5 pN, Sox2 and DNA underwent continued condensation, reducing the length of free DNA and bringing the two beads closer to each other (Fig. 3e). In contrast, a 10-pN force clamp largely abolished the condensation process (Fig. 3f), consistent with the above passive-mode results reporting a

7-pN maximum force that Sox2:DNA co-condensation can generate.

Next, we asked how much force is required to dissolve Sox2:DNA co-condensates. To address this question, we first formed Sox2 foci on a DNA tether under a low force (0.5 pN) and then gradually pulled the two beads apart, thereby increasing the force applied to the tether (Fig. 3g). From the resultant force-extension curve, we found that the extension of a Sox2-bound tether was much shorter than that of a bare DNA tether, indicating significant DNA accumulation inside the condensates (Fig. 3h). Some transitions were observed in the force-extension curve, which likely correspond to force-induced condensate dissolution events (Fig. 3h). Nonetheless, a significant fraction of condensates persisted even when the force reached the DNA overstretching regime (~65 pN), as reflected by the shorter extension of Sox2-bound DNA at high forces compared to bare DNA (Fig. 3h). Concomitant fluorescence imaging confirmed that many Sox2 foci remained intact during pulling (Fig. 3i). These results demonstrate that Sox2:DNA co-condensates are stable and resistant to high disruptive forces.

The optical tweezers results corroborate the surface-based TIRFM results, together revealing the mechanical effects of Sox2:DNA co-condensation. However, we note that a force on the order of 7 pN by itself is not sufficient to break intact DNA. Indeed, we observed few breakage events in the optical tweezers experiments. This discrepancy can be attributed to 1) the fact that we did not use any intercalating dye to stain DNA in the optical tweezers experiments; and 2) different illumination geometries between the TIRFM and optical tweezers assays, which may render DNA in the latter assay less susceptible to nicks and other types of photodamage. We stress that the force values obtained from the optical tweezers assay represent a more direct and accurate measure of the mechanical tension that Sox2 condensates exert on DNA.

**Nucleosomes attenuate the mechanical stress that Sox2 condensation exerts on DNA.** Given that Sox2 is a nucleosome-binding pioneer TF[24], we asked how the mechanical stress exerted by Sox2:DNA co-condensation on DNA may be regulated by nucleosome wrapping and chromatin organization. To this end, we loaded histone octamers containing Cy3-labeled H2B onto surface-immobilized λDNA in the TIRFM setup (Fig. 4a) and then added Cy5-Sox2 to bind the nucleosomal DNA (Fig. 4b). As expected, we observed that Sox2 foci nucleate around nucleosome locations (Fig. 4c, d). Sox2 foci preferentially colocalized with nucleosomes over bare DNA sites (Fig. 4e, f). The majority of Sox2:nucleosome foci contained multiple Sox2 molecules based on the Cy5 fluorescence intensity, similar to the Sox2 foci on bare DNA (Supplementary Fig. 9a).

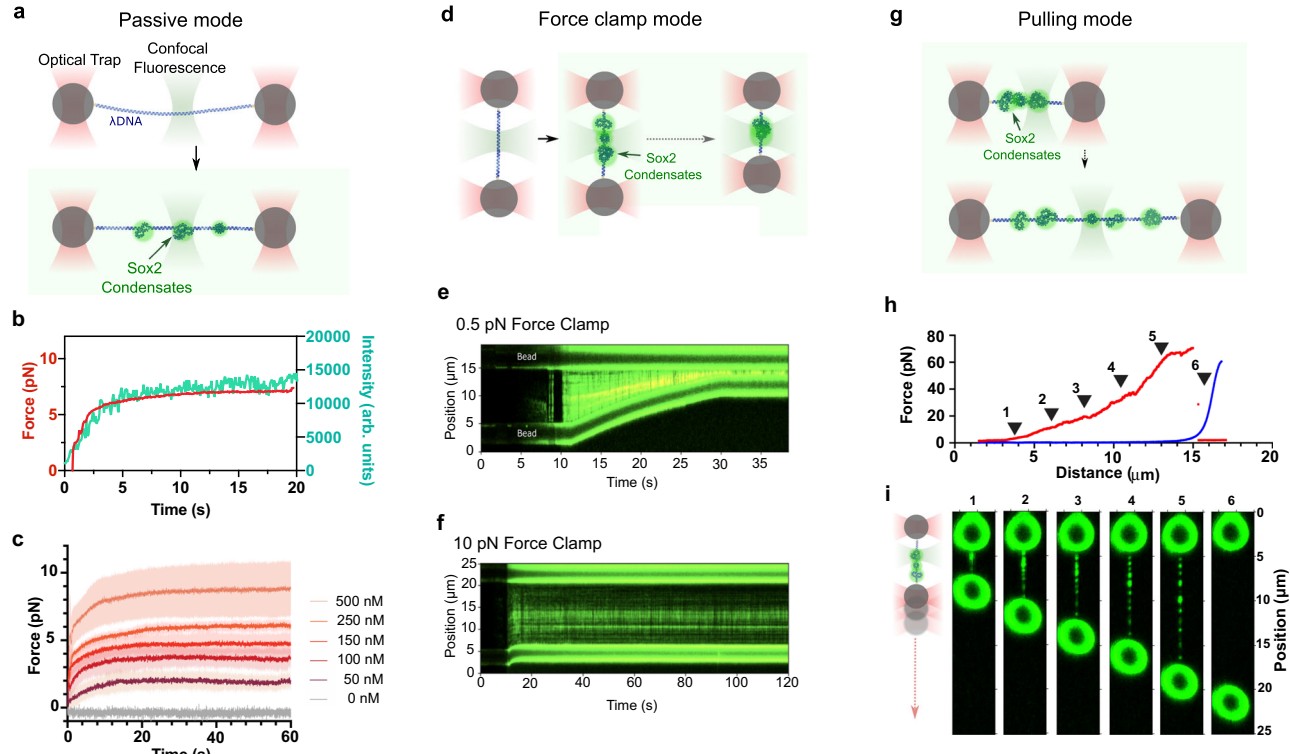

**Fig. 3 Optical tweezers assays for quantitative measurements of the force generated by Sox2:DNA co-condensation. a** Schematic of the optical tweezers assay that measures force generated by Sox2-mediated DNA condensation. The trap separation was fixed in this experiment. **b** Force measurements (red) and the corresponding fluorescence intensities (green) as a function of time for the assay depicted in **a**. Data are averaged from 4 representative tethers. **c** Force measurements made with different Sox2 concentrations. The colored lines correspond to the mean forces as a function of time averaged from multiple DNA tethers ($n = 4$ or 5). The shades correspond to standard deviation. **d** Schematic of force-clamp experiments. The force applied to the tether was kept at a constant value via feedback such that DNA condensation would result in shortening of the tether. **e** A representative kymograph showing significant tether contraction and Sox2 condensate formation under a 0.5-pN force clamp. **f** A representative kymograph showing suppressed tether contraction under a 10-pN force clamp. **g** Schematic of pulling experiments. Sox2 condensates first formed on tethered λDNA under a low force (~0.5 pN). The tether was then subjected to mechanical pulling by gradually separating the two traps apart. **h** A representative force-distance curve from pulling a λDNA tether harboring Sox2 condensates (red line) in comparison to a representative curve from pulling a bare λDNA (blue line). The black arrowheads denote selected time points imaged in **i**. **i** Two-dimensional fluorescence scan of the same tether as in **h** (red line) at selected time points during pulling showing that Sox2 condensates persisted under forces up to 60 pN (time points #1–5) until tether rupture (time point #6). Source data are provided as Source Data Fig. 3.

Strikingly, we detected drastically fewer DNA breakage events upon the formation of Sox2 foci on nucleosomal DNA than on bare DNA (Fig. 4g). In the few examples in which nucleosomal DNA breakage did occur, the tether ruptured at one of the anchor positions, and the full DNA contour was sustained and underwent rigid-body-like fluctuations (Supplementary Fig. 9b and Supplementary Movie 5). This is in contrast to the breakage events observed on bare DNA where the tether broke in the middle and the Sox2/DNA signals abruptly collapsed into the two anchor positions (Fig. 2a and Supplementary Movie 2). We also analyzed the single-tethered nucleosomal λDNA molecules and found that the addition of Sox2 did not significantly suppress their fluctuating motions (Fig. 4h–j and Supplementary Movie 6), again in contrast to the bare DNA results (Fig. 1i and Supplementary Movie 1).

Finally, we performed optical tweezers experiments to directly measure forces exerted by Sox2 condensates on nucleosomal DNA. We assembled histone octamers containing AlexaFluor488-labeled H2A onto a λDNA tether, moved the tether to a channel containing Cy3-Sox2, and monitored the force reading in the passive mode (Fig. 5a). Satisfyingly, we observed that, even though the Sox2 foci predominantly colocalized with nucleosomes (Fig. 5b), their formation hardly caused any increase in force, in contrast to the significant force increase on bare DNA (Fig. 5c).

These results corroborate the above TIRFM data, together suggesting that nucleosomes attract Sox2 proteins and attenuate the force exerted by Sox2 condensates on DNA (Fig. 6).

## Discussion

Prior to this study, the forces generated by co-condensation between DNA and proteins—such as FoxA1 and PARP1—were estimated to be on the order of sub-pN, placing them among the weakest cellular forces alongside those generated by loop-extruding SMC complexes such as condensin and cohesin[9,13,25]. Here we show that Sox2, an abundant TF central to pluripotency and embryogenesis, can actively generate condensation forces up to 7 pN, one order of magnitude higher than previously reported values. The cellular Sox2 concentration is estimated to be in the low micromolar range[26,27]. Therefore, we speculate that the forces generated by Sox2 in vivo are at least comparable to those measured in our in vitro experiments. It is worth noting that Klf4, another pluripotency TF, can also form condensates on DNA against a relatively high force (~8 pN)[28]. Once formed, the Sox2:DNA co-condensates are extremely stable, resistant to pulling forces sufficient to overstretch B-form DNA. In comparison, a fraction of the condensates formed by DNA and Heterochromatin Protein 1α can resist disruptive forces of up to

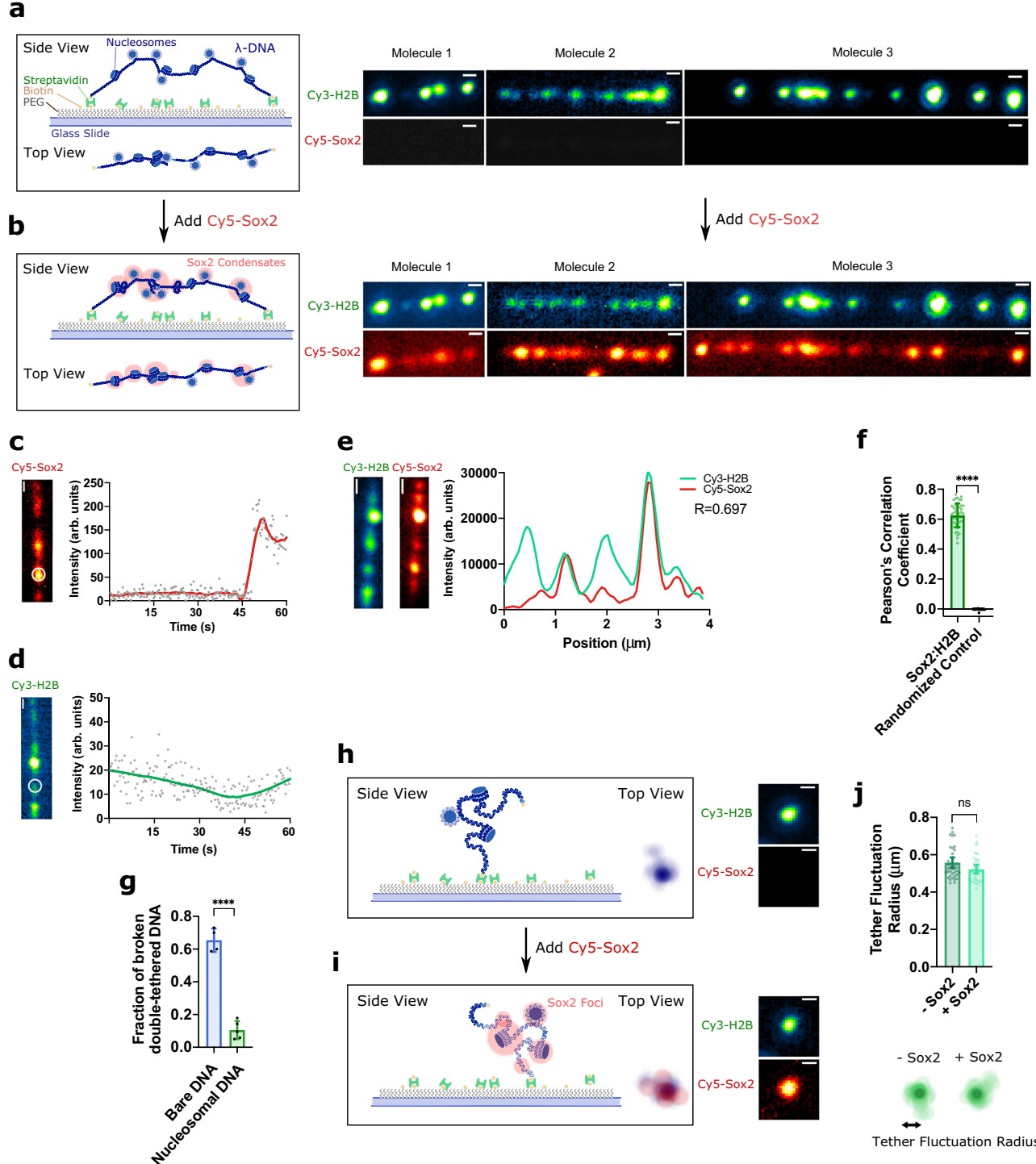

40 pN[29]. These findings are of significance because they show that forces exerted by certain protein:DNA co-condensates are comparable to other cellular forces such as those generated by molecular motors[30]. Therefore, TF-mediated DNA condensation can potentially impact other genomic processes by maintaining mechanically stable foci. This represents an additional, but not mutually exclusive, mechanism for gene regulation besides the canonical sequence-specific TF-DNA interaction paradigm.

The critical force below which a protein-rich condensate is able to pull DNA inside likely depends on the physicochemical properties of the condensate, such as its surface tension[31], which in turn are determined by the characteristics of the TF including its charge distribution and intrinsic disorder. Here, we show that the ability of Sox2 to generate high forces through co-condensation with DNA critically relies on its IDRs. On the other hand, Sox2's DNA-binding HMGB domain alone is sufficient for forming co-condensates with DNA, reminiscent of recent findings with Klf4 and SMN proteins[32,33]. It has been reported that Sox2 can form a dimer on DNA that requires a motif located at the C-terminus of HMGB[34]. Hence the dimerization activity of Sox2-HMGB may underlie its ability to form co-condensates with DNA, whereas the multivalent interaction mediated by Sox2's IDRs is likely responsible for its force-generating ability. We also observed that the generation of high

**Fig. 4 Nucleosomes colocalize with Sox2 condensates and attenuate their mechanical effects on DNA. a** (Left) Schematic of double-tethered λDNA loaded with nucleosomes. (Right) Time-averaged projections of three double-tethered nucleosomal DNA molecules with different end-to-end distances (among 5 independent experiments). Nucleosomes were visualized by Cy3-labeled histone H2B. Scale bar, 0.5 μm. **b** Schematic and time-averaged projections of the same three nucleosomal DNA molecules as in **a** when incubated with 10 nM Cy5-labeled Sox2. Scale bar, 0.5 μm. **c** Real-time tracking of Cy5-Sox2 intensities at a nucleosome position (circled region) on a double-tethered nucleosomal λDNA. Scale bar, 0.5 μm. **d**, Corresponding Cy3-H2B intensities within the same circled region as in **c**. Scale bar, 0.5 μm. **e** (Left) Snapshot of a double-tethered DNA harboring Cy3-H2B nucleosomes and Cy5-Sox2 condensates. (Right) Intensity profiles of Cy3-H2B (green) and Cy5-Sox2 (red) along the length of the same DNA molecule. *R* value represents Pearson's correlation coefficient. Scale bar, 0.5 μm. **f** Pearson's correlation coefficients averaged from all aligned Cy3-H2B and Cy5-Sox2 intensity profiles and from Costes' randomized control ($n = 158$). Error bars denote 95% CI. Significance was obtained using an unpaired two-sample *t* test (****$P < 0.0001$). **g** Fraction of double-tethered bare DNA ($n = 379$) versus nucleosomal DNA molecules ($n = 303$) that broke after 15 min of incubation with 10 nM Sox2. Data are averaged from at least three fields of view. Error bars denote standard deviation. Significance was obtained using an unpaired two-sample *t* test (****$P < 0.0001$). **h** Schematic and time-averaged projection of a single-tethered nucleosomal λDNA visualized by Cy3-H2B fluorescence. Scale bar, 0.5 μm. **i** Schematic and time-averaged projection of the same nucleosomal DNA molecule as in **h** when incubated with 10 nM Cy5-Sox2. Scale bar, 0.5 μm. **j** Average fluctuation radius of single-tethered nucleosomal λDNA in the absence or presence of Sox2 ($n = 42$). Error bars denote 95% CI. Significance was obtained using an unpaired two-sample *t* test (ns, $P = 0.28$). Source data are provided as Source Data Fig. 4.

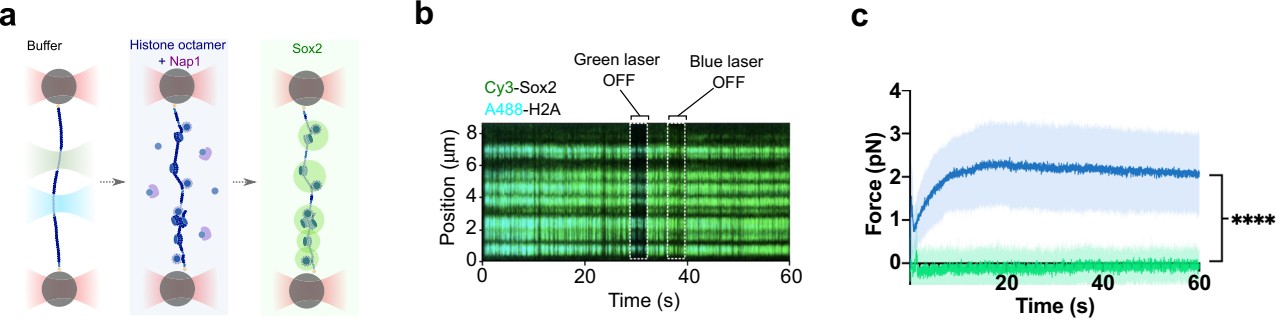

**Fig. 5 Nucleosomes attenuate the condensation force exerted by Sox2 on DNA. a** Schematic of in situ nucleosome assembly and Sox2 condensate formation on a λDNA tether in an optical tweezers assay. **b** A representative kymograph showing the colocalization of Sox2 condensates (green) with nucleosomes (cyan) on a λDNA tether. **c** Force measurements on nucleosomal DNA as a function of time (green line) for the assay depicted in **a**. Data are averaged from 7 representative tethers. Force measurements on bare DNA tethers are shown in blue (averaged from 14 tethers). The dark colored lines correspond to average force trajectories. The shades correspond to standard deviation. The Sox2 concentration in these experiments was 75 nM. Significance was obtained using an unpaired two-sample *t* test (****$P < 0.0001$). Source data are provided as Source Data Fig. 5.

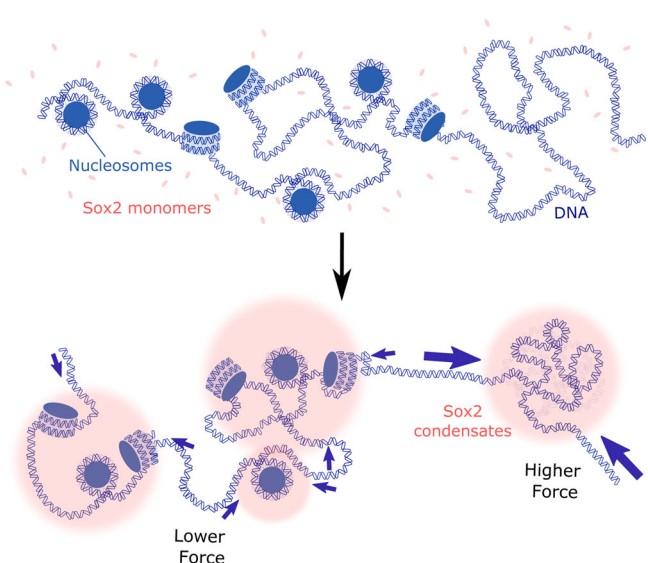

**Fig. 6 Working model.** Sox2 condensates exert differential mechanical tension within nucleosome-occupied genomic regions versus nucleosome-free regions. Nucleosomes can thus serve as mechanical sinks to regulate the force generated by TF:DNA co-condensation.

forces by Sox2 condensates, signified by the DNA breakage events, predominantly occurs early but diminishes as the condensates' mobility decreases over time. This observation indicates that the maturation of Sox2:DNA co-condensates from a liquid-like form to a solid-like one—akin to what was described in other systems[35]—attenuates their force-generating capacity.

Eukaryotic chromatin is known to alter the mechanics of DNA by changing its persistence length and torsional stiffness[36,37]. Our work presented here adds to the mechanical regulatory roles of chromatin by showing that nucleosomes sequester TFs such as Sox2 from exerting high mechanical tension on DNA. Considering that Sox2 has the inherent ability to bind nucleosomes, it is conceivable that such interactions serve as nucleation events that eventually recruit most of the available Sox2 molecules to nucleosomal sites. Due to the strong contacts made between DNA and the histone octamer, Sox2 condensation on nucleosomal sites may not be able to reel in additional free DNA. As such, these Sox2:nucleosome condensates serve as "mechanical sinks" that buffer the stress accumulated within genomic DNA, thereby protecting chromosomes from potentially deleterious nuclear forces. Whether this effect is limited to pioneer transcription factors is an interesting question to pursue in the future[38].

In sum, our study highlights the mechanical impact of TF:DNA co-condensation on the DNA and the role of chromatin in regulating such impact. It can be envisioned that the nucleosome

landscape—shaped by many factors and altered during development and disease—is directly related to the force field in the nucleus. Further studies are warranted to elucidate this relationship, which will improve our understanding of how chromatin-based mechanical events influence genome integrity and function[39].

## Methods

### Protein purification and labeling

*Sox2*. Human Sox2 proteins were expressed and purified as previously described[16]. In brief, Sox2-FL and Sox2-HMGB constructs were cloned into the pET28B plasmid, expressed in Rosetta (DE3) plyS cells (Novagen #70956-3) in LB media at 37 °C until reaching an $OD_{600}$ of ~0.6, and induced with 0.5 mM IPTG at 30 °C for 2 h. Cells were harvested, lysed, and purified using a Ni-NTA affinity column under denaturing conditions. Eluted Sox2 was refolded by changing to a zero-urea buffer using a desalting column (GE healthcare #17-1408-01). Further purification was performed by gel filtration on a Superdex 200 10/300 GL column (GE Healthcare). Fluorescence labeling was performed as previously described[16]. In brief, Cy5 or Cy3 maleimide (GE healthcare) was mixed with Sox2 at a molar ratio of ~2:1. For Sox2-FL, the dye was conjugated to the only native cysteine C265. For Sox2-HMGB, a K42C mutation was introduced by site-specific mutagenesis. Free dye was removed by gel filtration on a Superdex 200 10/300 GL column.

*Histone octamer*. Recombinant histone octamers from *Xenopus laevis* were purified and labeled as previously described[16]. In brief, each of the four core histones (H2A, H2B, H3 and H4) was individually expressed in BL21 (DE3) cells, extracted from inclusion bodies, and purified under denaturing conditions using Q or SP ion exchange columns (GE Healthcare). Octamers were refolded by dialysis and purified by gel filtration on a Superdex 200 10/300 GL column. To label the octamer, single-cysteine constructs H2B T49C and H2A K120C were generated by site-directed mutagenesis and incubated with Cy3 and A488 maleimide at 1:5 molar ratio, respectively.

*Linker histone H1*. His-Sumo-H1.4$^{A4C}$-GyrA-His was expressed and purified as described previously[40] with minor adjustments. Briefly, the construct was expressed in Rosetta DE3 cells overnight at 16 °C. Cells were lysed and lysate incubated with Ni-NTA beads (Bio-Rad). 1 mM DTT was added to the eluent, and it was incubated with Ulp-1 (1:100 v/v) for 1 h at room temperature. Following this, 500 mM β-mercaptoethanol was added. The mixture was run on a Hi-Trap SP column, and fractions containing full-length H1.4$^{A4C}$ were pooled and injected on a semi-preparative HPLC C18 column. Pure fractions of H1.4$^{A4C}$ were pooled and lyophilized. Lyophilized H1.4$^{A4C}$ was resuspended in H1 labeling buffer (6 M Guanidine, 20 mM Tris pH 7.5, 0.2 mM TCEP). It was mixed with 3 molar equivalents of Cy3 maleimide for 1 h at room temperature, followed by quenching with 1 mM β-mercaptoethanol. This was injected on a semi-preparative HPLC C18 column. Pure fractions of Cy3-H1.4 were pooled and lyophilized. Cy3-H1.4 was resuspended in buffer (20 mM Tris pH 7.5, 200 mM NaCl) before use.

### Single-molecule TIRFM experiments.
Single-molecule imaging was conducted on a total-internal-reflection fluorescence microscope (Olympus IX83 cellTIRF) and visualized using Metamorph v7.8 software. PEG slides were prepared as previously described[16]. The assembled flow chamber was infused with 20 μL of 0.2 mg/mL streptavidin (Thermo Fisher Scientific), incubated for 5 min, and washed with 250 μL of T150 buffer (50 mM Tris pH 7.5, 150 mM NaCl, 0.0075% Tween). Biotinylated λDNA (LUMICKS) was immobilized by slowly injecting a diluted 10–20 pM solution at a volume of 40–80 μL over the course of 2 min. Afterwards, 250 μL of T150 buffer was flowed into the chamber to wash away molecules that were not immobilized. For T4 ligase treatment, 20 μL of 1:20 diluted T4 ligase (NEB) in T4 ligase buffer was flowed into the chamber, incubated for 10 min, and washed away with 250 μL of T150 buffer. 100 μL of T150 buffer containing YOPRO1 (20 nM unless specified otherwise) and an oxygen scavenging system (4% w/v glucose, 1.5 mg/mL glucose oxidase, 0.072 mg/mL catalase, 2 mM Trolox) was then flowed in to visualize immobilized λDNA.

In nucleosome experiments, we adopted a previously described protocol with minor modifications[41]. In brief, in situ nucleosome formation was achieved by flowing in 15 nM of Cy3-labeled histone octamer and 30 nM of Nap1 in T150 buffer into the chamber followed by a 5-min incubation. The chamber was then flushed with 250 μL of T150 to wash away any free histone octamer and Nap1.

A solution containing a specified concentration of Cy5-labeled Sox2 and the above imaging buffer (i.e., T150, YOPRO1, and oxygen scavenging system) was prepared, 50 μL of which was flowed into the microfluidic chamber, and movies/images were recorded. H1 imaging was similarly performed with a specified concentration of Cy3-H1. Movies were recorded at room temperature with a frame rate of 300 ms. 488-nm, 532-nm, and 640-nm lasers were used to excite YOPRO1, Cy3, and Cy5/TOTO3 dyes, respectively. Movies were subsequently displayed and analyzed using plugins in ImageJ/FIJI.

## TIRFM data analysis

*Analysis of DNA envelope width and fluctuation radius*. We followed an analysis pipeline as previously described[13]. In brief, time-averaged projections of DNA images were generated in conditions with/without proteins. Transverse line profile of the DNA intensity was generated by drawing a line perpendicular to the middle of the DNA, which gives the maximum DNA width. Background was subtracted off these profiles, and a Gaussian curve was fitted to each line profile. The DNA envelope width and fluctuation radius were defined as two times the standard deviation of the fitted Gaussian curve.

*Estimation of DNA content and Sox2 counts in a cluster*. The YOPRO1 intensity profile was extracted and background subtracted. The estimated DNA content within each cluster was calculated as similarly described[42] and shown below:

$$DNA\ content\ (bp) = \frac{DNA\ intensity\ in\ cluster \times 48,502\ (\lambda DNA\ length)}{Total\ DNA\ intensity} \quad (1)$$

To estimate the number of Sox2 molecules in each cluster, we extracted the Cy5 intensity profile that colocalized with λDNA after subtraction of background signals. We then extracted the intensity profiles of Cy5-Sox2 non-specifically adsorbed to surface in the same field of view, which we assumed as monomers. The number of Sox2 molecules within each cluster on λDNA was calculated by dividing the integrated cluster intensity by the monomer intensity.

*Colocalization analysis*. Time-averaged projection of the images in each fluorescence channel was generated, and background was subtracted. The regions of interest were segmented and extracted for further analysis. Pearson's correlation coefficients in each condition were calculated using JaCoP plugin in FIJI[43]. Costes' randomized control, which describes the correlation between randomly shuffled pixels of two compared images[44], was also calculated using the JaCoP plugin.

*Condensation time analysis*. Each immobilized λDNA molecule in a field of view was individually monitored, and the time when a molecule condensed was defined as the transition at which the molecule completely lost slack/fluctuations. We subsequently ranked the condensation times and recorded the 75th and 25th percentile values ($T_{75}$ and $T_{25}$, respectively). The average condensation time ($T_{condense}$) was calculated as $T_{75}$-$T_{25}$.

*Mobility analysis*. Kymographs were extracted from TIRF microscopy movies using the kymographBuilder plugin in Fiji. The Sox2 foci were then manually extracted using the Kymotracker 'greedy' tracking algorithm[45,46]. Early and late Sox2 condensate events were recorded from movies taken ~5 min and ~15 min after Sox2 injection, respectively. One-dimensional mean squared displacement (MSD) was then applied using a maximum delay time of 4 s (0.3 s time steps) using a custom python script written based on the description and methods from the @msdanalyzer MATLAB per-value class[47]. A Savitzky-Golay filter (third order polynomial with an eleven-frame window) was applied to smooth traces in preparation for MSD analysis. Diffusion coefficients were only calculated if the coefficient of determination ($R^2$) of the linear fit was greater than 0.8. Approximately 67% of early Sox2 traces and 50% of late Sox2 traces met the required parameters for diffusion coefficient fitting.

### Optical tweezers experiments.
Single-molecule optical tweezers experiments were performed at room temperature on a LUMICKS C-trap combining confocal fluorescence microscopy and dual-trap optical tweezers as previously described[23]. In brief, we trapped two streptavidin-coated polystyrene beads (Spherotech) with a 1064-nm trapping laser and moved these beads to a channel containing biotinylated λDNA (LUMICKS). Single DNA tethers were selected based on the force-extension curve. The DNA tether was then moved into a channel containing Cy3-labeled Sox2 in T150 buffer. Cy3-Sox2 on DNA was visualized by confocal scanning with a 532-nm excitation laser. Correlative force and fluorescence measurements were made under different operation modes (force clamp mode, passive mode, or pulling mode)[48] as specified in the figure legends.

Nucleosomal DNA experiments were similarly performed. To assemble nucleosomes in situ, a single λDNA tether was moved into a channel containing 12 nM of A488-H2A histone octamer and 48 nM Nap1 in HR buffer (30 mM Tris-OAc pH 7.5, 20 mM Mg(OAc)$_2$, 50 mM KCl, 1 mM DTT, 40 μg/mL BSA), and incubated at a fixed trap distance of 10 μm for 20 s under flow and another 20 s without flow. The tether was then moved into another channel containing 0.5 mg/mL salmon sperm DNA in HR buffer, in which a flow was applied for 30 s to remove free histones and Nap1. Before moving to the Sox2 protein channel, the force was reset to zero to remove any influence of nucleosome wrapping on the force reading. Force and fluorescence data were generated via Bluelake software v2.1.5 (LUMICKS) and processed using a custom GUI Python script titled "C-Trap.h5 File Visualization GUI" (https://harbor.lumicks.com/single-script/c5b103a4-0804-4b06-95d3-20a08d65768f).

### Electrophoretic mobility shift assay (EMSA).
DNA substrate was prepared via PCR and gel extraction of a 233-bp construct containing the Sox2 binding motif engineered into a 601 sequence (Supplementary Table 1) as previously described[16].

10 nM of DNA substrate was incubated with Sox2 and HMGB constructs in T150 buffer at room temperature for 30 min. The reaction mixture was loaded onto a 5% non-denaturing polyacrylamide gel, which was run in 0.5× Tris-Borate-EDTA at 4 °C at 100 V for 90 min, stained with SYBR Gold (Invitrogen), and visualized using a Typhoon FLA7000 gel imager (GE Healthcare).

**Statistics and reproducibility.** Statistical tests and *P* values were reported in the figure legends (ns, not significant; *$P < 0.05$, **$P < 0.01$, ***$P < 0.001$, ****$P < 0.0001$). All experiments were independently repeated at least three times with similar results. Representative results are shown in figures.

**Reporting summary.** Further information on research design is available in the Nature Research Reporting Summary linked to this article.

## Data availability
The data that support this study are available from the corresponding author upon reasonable request. Source data are provided with this paper.

## Code availability
All specified scripts used to process and analyze C-trap experiments can be accessed on LUMICKS Harbor site (https://harbor.lumicks.com). All custom-written codes can be made available upon reasonable request.

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

## Acknowledgements
We thank members of the Liu Laboratory for discussions, R. Shih and the O'Donnell Laboratory (Rockefeller University) for the Nap1 protein, and Rockefeller University Bio-Imaging Resource Center for help with image analysis. T.N. and J.C. are supported by a Medical Scientist Training Program grant from the National Institute of General Medical Sciences of the National Institutes of Health under award number T32GM007739 to the Weill Cornell/Rockefeller/Sloan Kettering Tri-Institutional MD-PhD Program. Y.D. is supported by NIH grant R35GM138386 and CCSG core grant P30CA008748. S.Liu is supported by the Robertson Foundation, the Pershing Square Sohn Cancer Research Alliance, and an NIH Director's New Innovator Award (DP2HG010510).

## Author contributions

S.Liu and T.N. conceived the project and designed the experiments. S.Li cloned the Sox2 constructs and prepared the histone octamers. T.N. purified and labeled Sox2 proteins, performed TIRFM experiments, and performed bulk assays with help from H.N. J.C. performed the optical tweezers experiments. J.W. wrote scripts for data analysis. A.O. and Y.D. provided the linker histones. T.N. and S.Liu wrote the manuscript with input from all authors.

## Competing interests

The authors declare no competing interests.

## Additional information

**Peer review information** *Nature Communications* thanks other anonymous reviewer(s) to the peer review of this work. Peer review reports are available.

