## [Peer Review File · Nature Communications]

REVIEWER COMMENTS

Reviewer #1 (Remarks to the Author):

This manuscript addresses an interesting observation that Sox2, a nucleosome-binding pioneer forms co-condensates with DNA. Using single-molecule fluorescence assays, the authors explore the capabilities of condensation of Sox2 along with a DNA. The observation that Sox2:DNA co-condensate exerts high tension on DNA is interesting and timely because Sox2 is a very important factor and is not well understood. Therefore, this study can contribute to understanding how TFs interact with nucleosomes for further transcription processes.

However, I found a possibility of improper interpretation on a sudden DNA breakage, induced by Sox2:DNA condensation. Because this interpretation can mislead readers, I ask the authors to carefully address the points regarding the DNA breakage (see below). Except the point, I would like to support publishing this work in Nature Communications.

Major comments

1. The interpretation of the breakage experiment result is not logical. In Fig. 3B, the authors showed that the condensation induces force up to 7 pN. However, 7 pN should not be enough to break the covalent bonding between DNA base pairs. Although the authors tried to show reasonable control data that photochemical reaction is not enough to induce significant DNA breakage, however, DNA breakage (~ 2 %) was observed (Fig. 2B), meaning that the photo-chemical reaction might already slightly destabilize the DNA. Besides, the intercalated-DNA breakage upon laser excitation is easily broken is a well-known phenomenon. Hence, the DNA breakage results could be the combinatory effect of both the photo-chemical reaction of DNA and mechanical tension. Therefore, at this point, it is difficult to rule out other scenarios because 7 pN is too low to break a covalent bonding. For example, if Sox2 induces protein-induced fluorescence enhancement (PIFE) of YoYo dyes (10.1073/pnas.1017672108), then direct interaction between Sox2 and DNA would increase the photochemical reaction of YoYo dyes on DNA. Thus, the authors need to show whether Sox2-condensation induces DNA breakage on label-free DNA. Perhaps, the authors can perform optical-tweezer experiments for DNA breakage experiment using unlabeled DNA.

2. Regarding comment 1, I would recommend not to show the fraction of broken double-tethered DNA (Fig. 2B). Instead, the optical tweezer data shown in Fig. 3 would be enough to address that high tension is generated by Sox2-condensate. Moreover, Fig. 4G also needs to be removed, and instead, it would be good to show the force when Sox2 condensates along nucleosomes as in Fig.3B. In addition, I expect lower forces (< 7 pN) would be observed in the presence of nucleosomes. This control experiment is

important to rule out the possibility that nucleosomes attenuate the direct interaction between Sox2 and DNA that can just simply decrease the photochemical effect of YoYo1.

Minor comments

1. (Suggestion) It would be good to show a force-distance curve with Sox2-condensation in the presence of nucleosome to quantitatively show the attenuation effect of the nucleosome in Fig. 3D.
2. The scale bars need to be corrected in Fig. 1G, Fig. 4A, I, FigS5B, and Fig6A,B. They are located at very edge positions.

Reviewer #2 (Remarks to the Author):

The manuscript by Tuan Nguyen et al. uses single molecule techniques to show that Sox2 forms co-condensates with DNA and exerts mechanical tension on the DNA. The data to support this is overall convincing, but the authors need to verify this observation is not concentration dependent (see comments below). The authors go on to include nucleosomes in one experiment to show that they prevent this mechanical stress from occurring by Sox2. Overall, this paper is of interest to the readers of nature communications. Specific comments are listed below.

1. Did the authors do varying concentration of sox2 in their assay to determine if this is a concentration dependent phenomena.
2. Do the authors know how many sox2 molecules are within these condensates?
3. Is the end breakage biologically relevant or just an example of the tension that Sox2 puts on the DNA? Is this concentration dependent, and if so, will this occur within a cellular setting?
4. The authors perform a control experiment with Cy3-H1 to determine if the condensation occurs with other DNA binding proteins. They perform this experiment at 150pM H1 and don't see as much condensation. However, with the sox2 they do these same experiments at 10nM. This raises concern that the results are arising from these experiments being performed at substantially different concentrations. Did the authors test H1 at 10nM? The results in supplemental figure 5 indicate that H1 can also exert force on the DNA and it may be at a similar level as Sox2 if performed at the same concentration.
5. Does sox2 dimerize with itself and therefore promote formation of these condensations through the IDRs?

6. The authors need to describe how they generated the histone samples for the single molecule experiments in more detail. They describe the expression and labeling, but not how they formed the final DNA/nucleosome substrate. They indicate that Nap1 is flowed in with the histone octamer to generate the DNA/nucleosome substrate. Did the authors verify if Nap1 is still present in the sample chamber when Sox2 was added?

Reviewer #3 (Remarks to the Author):

In “Chromatin sequesters pioneer transcription factor Sox2 from exerting force on DNA”, Nguyen et al. present single-molecule studies of transcription factor Sox2 condensation on DNA.

The authors use single-molecule fluorescence microscopy on tethered DNA to image Sox2 assembly into condensates on DNA, which reduces transverse fluctuations in DNA and destabilizes DNA to the point of breakage. This is presented as evidence for Sox2 condensates generating large forces. At steady-state, Sox2 condensates fuse and split—on the same DNA strand and between different DNA molecules—which confirms liquid-like behavior. Two Sox2 mutants, one without an intrinsically disordered region and one with a mutation in DNA-binding domain, are shown to slow phase-separation and reduce Sox2-induced DNA breakage. Using optical traps with fluorescence microscopy, the authors measure a maximum force of 7 pN generated upon Sox2 condensation, an order of magnitude higher than that reported for other protein-DNA condensates. Sox2 condensates remain stably associated with the DNA even under mechanical tension high enough to unwind duplex DNA. Lastly, the authors present measurements of Sox2 on DNA pre-loaded with nucleosomes, showing that Sox2 colocalizes with nucleosomes, and their presence reduces Sox2-induced DNA breakage. These findings suggest that nucleosomes sequester Sox2 from exerting high forces on the genome, thereby resembling a mechanical “sink” that buffers stress within genomic DNA.

Overall the authors present an interesting finding which should be of broad interest to the scientific community, particularly researchers working on biomolecular condensates. The results are clearly and succinctly presented. The single-molecule fluorescence and optical tweezers experiments are first rate, and extract meaningful and novel physical properties of Sox2:DNA condensates. However, there are a number of important issues that need to be addressed. In particular, there are questions on the arguments for force generation and concerns over apparent inconsistencies between the fluorescence and optical tweezers measurements and their interpretation. The authors may be able to address these points and others (enumerated below) through additional measurements, including suggested controls. These questions need to be answered before a recommendation to publish can be made.

Major points:

The authors should present control experiments with unlabeled, wild-type Sox2. In the TIRFM and trap assays, other readouts (e.g. YOPRO1 signal, force) can be used to detect DNA condensation, breakage, and/or force generation, i.e. independent of Sox2 labeling. One should expect these behaviors to be reproduced with unlabeled Sox2. This control is critical to determine if Cy5 labeling affects Sox2 activity and to rule out any photophysical effects.

The authors argue that force generation by Sox2 condensates is responsible for DNA breakage, (observed by fluorescence imaging, Fig. 2). However, this mechanism does not seem plausible given the maximum condensation force of 7 pN (measured by optical trap, Fig. 3). It is unlikely that a force of 7 pN is sufficient to rupture duplex DNA. One possibility is that this force breaks the biotin-streptavidin linkages, but this is inconsistent with the observation of two fluorescence spots at the two tethering points after rupture (Fig. 2A). Another is that the DNA is highly nicked; in this case, 7 pN and thermal melting could perhaps lead to breakage, although this could depend highly on the density of nicks. In either case, one would expect breakage to occur in the optical trap measurements; yet it is curious that the authors do not report this. Do they observe breakage in the trap? If not, this suggests that another mechanism is at play, or that the different conditions between TIRFM and trap measurements could have an effect (e.g. labeled vs. unlabeled DNA, surface bound DNA vs. levitated DNA). The authors should address this point at length and present a plausible mechanism for DNA rupture. Perhaps performing optical trapping experiments with the two mutants could help in establishing a mechanism.

On a related point comparing the TIRFM and trap measurements, the authors should be able to estimate the tension on the DNA from its transverse fluctuations as measured in the TIRFM experiments (Fig. 1E-F). How do these numbers compare to the forces measurement by optical tweezers?

In the TIRFM measurements, the authors should present data on the effect of the surface on the condensate mobility. Is it possible that the decrease in transverse fluctuations in DNA is not due to increased tension, but due to the Sox2 condensates sticking to the surface and providing additional attachment points to the DNA? The authors could compare condensate mobility (e.g. diffusion constant) in the surface-based TIRFM measurements vs. solution-based trap measurements.

Some protein-NA condensates have been reported to undergo a slow maturation where they transition to a more solid-like behavior. Are there any reports of this behavior for Sox2:DNA condensates? Do the authors see any evidence for maturation in their data (e.g. decreased condensate fusion events and overall mobility over time)? It may be that condensate maturation alters the strength of Sox2 binding to DNA, and perhaps this could result in breakage. The time frame for DNA breakage is in line with those of maturation in other systems.

Specific points:

The authors claim to measure the DNA content of the condensates (e.g. Fig. 1C), but how these values are determined is not well explained in Materials and Methods (see “Estimation of DNA content and Sox2 counts in a cluster”). The procedure outlined gives the number of Sox2 monomers, not the DNA content. Is the DNA content assumed to be equal to: (binding site size) x (# of monomers)? Is the DNA footprint of Sox2 well known? This should be clarified

The authors show that Sox2 binds DNA sequence-specifically (SI Fig. 2), but do not do much with this information. Does the position distribution of the condensates from fluorescence imaging (e.g. Fig. 1C or 3E) correlate with DNA sequence?

On a related point, in Fig. 4E-F the authors show a correlation analysis for nucleosome and Sox2 condensate position. This is compared to a randomized control, but perhaps the data should be compared to Sox2 distribution on bare DNA, which is presumably not random based on the sequence specificity?

In addition:

1. For all figures, the authors should adjust the white scale bars such that they are all on a black background, not at the edge of the frames which makes them hard to see. Also, in the captions, they should describe what the arrows are indicating throughout (e.g. this is missing in Fig. 3D).

2. Consider adding the kymographs of SI Fig. 9 to Fig. 3.

3. In Fig. 3B, could the authors show the total fluorescence intensity measured at the same time? This would confirm that the tension increases as the condensates form. The authors could consider showing an average behavior instead of 1-4 representative traces (or show representative traces in grey and an average trace in color).

Point-by-point response to the reviewers' comments

Reviewer #1 (Remarks to the Author):

This manuscript addresses an interesting observation that Sox2, a nucleosome-binding pioneer forms co-condensates with DNA. Using single-molecule fluorescence assays, the authors explore the capabilities of condensation of Sox2 along with a DNA. The observation that Sox2:DNA co-condensate exerts high tension on DNA is interesting and timely because Sox2 is a very important factor and is not well understood. Therefore, this study can contribute to understanding how TFs interact with nucleosomes for further transcription processes.

However, I found a possibility of improper interpretation on a sudden DNA breakage, induced by Sox2:DNA condensation. Because this interpretation can mislead readers, I ask the authors to carefully address the points regarding the DNA breakage (see below). Except the point, I would like to support publishing this work in Nature Communications.

Response: We thank the reviewer for the positive assessment of our work. We have performed additional control experiments and expanded the discussion to address the reviewer's concerns with regards to the DNA breakage.

Major comments

1. The interpretation of the breakage experiment result is not logical. In Fig. 3B, the authors showed that the condensation induces force up to 7 pN. However, 7 pN should not be enough to break the covalent bonding between DNA base pairs. Although the authors tried to show reasonable control data that photochemical reaction is not enough to induce significant DNA breakage, however, DNA breakage (~ 2 %) was observed (Fig. 2B), meaning that the photochemical reaction might already slightly destabilize the DNA. Besides, the intercalated-DNA breakage upon laser excitation is easily broken is a well-known phenomenon. Hence, the DNA breakage results could be the combinatory effect of both the photo-chemical reaction of DNA and mechanical tension. Therefore, at this point, it is difficult to rule out other scenarios because 7 pN is too low to break a covalent bonding. For example, if Sox2 induces protein-induced fluorescence enhancement (PIFE) of YoYo dyes (10.1073/pnas.1017672108), then direct interaction between Sox2 and DNA would increase the photochemical reaction of YoYo dyes on DNA. Thus, the authors need to show whether Sox2-condensation induces DNA breakage on label-free DNA. Perhaps, the authors can perform optical-tweezer experiments for DNA breakage experiment using unlabeled DNA.

Response: We thank the reviewer for bringing up this important issue. We agree with the reviewer that ~7 pN of force by itself is too low to rupture intact DNA, and that DNA breakage is likely caused by the combinatory effect of the mechanical tension and the photo-chemical reaction of DNA. We performed additional control experiments to support this speculation. First, we washed out the YOPRO1 DNA-intercalating dye after the initial imaging of DNA and before the addition of Sox2 and found a much lower fraction of ruptured DNA (new Supplementary Fig. 5C). Second, we treated the λ DNA with T4 ligase to seal potential nicks and observed much fewer DNA breakage events upon Sox2 condensation (new Supplementary Fig. 5D). We also performed

optical tweezers experiments using unlabeled DNA as the reviewer suggested and observed few breakage events. We now interpret the DNA breakage results with the above caveats clearly stated in the revised manuscript.

2. Regarding comment 1, I would recommend not to show the fraction of broken double-tethered DNA (Fig. 2B). Instead, the optical tweezer data shown in Fig. 3 would be enough to address that high tension is generated by Sox2-condensate. Moreover, Fig. 4G also needs to be removed, and instead, it would be good to show the force when Sox2 condensates along nucleosomes as in Fig.3B. In addition, I expect lower forces (< 7 pN) would be observed in the presence of nucleosomes. This control experiment is important to rule out the possibility that nucleosomes attenuate the direct interaction between Sox2 and DNA that can just simply decrease the photochemical effect of YoYo1.

Response: We performed optical tweezers experiments to directly measure the force that Sox2 condensates exert on nucleosomal DNA without the DNA staining dye. The results are now shown in the new Fig. 5. It is gratifying to see that, consistent with what the TIRFM experiments suggested, the force on nucleosomal DNA is drastically lower than that on bare DNA. We thank the reviewer for suggesting this important experiment.

We decide to keep the DNA breakage results in the manuscript because we still think that they are useful for comparing the mechanical effect of different protein constructs, given that they were obtained under the same experimental conditions (laser power, dye concentration, etc.). We added a detailed discussion of the caveats in the revised manuscript.

Minor comments

1. (Suggestion) It would be good to show a force-distance curve with Sox2-condensation in the presence of nucleosome to quantitatively show the attenuation effect of the nucleosome in Fig. 3D.

Response: We added a new main figure (Fig. 5) to demonstrate the attenuation effect of nucleosomes on the condensation force generated by Sox2, including a cartoon of the experimental procedure, a representative kymograph showing the colocalization of Sox2 and nucleosomes, and the corresponding force curve that quantitatively measures the tension on nucleosomal DNA versus bare DNA.

2. The scale bars need to be corrected in Fig. 1G, Fig. 4A, I, Fig55B, and Fig6A,B. They are located at very edge positions.

Response: We moved the scale bars as suggested.

Reviewer #2 (Remarks to the Author):

The manuscript by Tuan Nguyen et al. uses single molecule techniques to show that Sox2 forms co-condensates with DNA and exerts mechanical tension on the DNA. The data to support this is

overall convincing, but the authors need to verify this observation is not concentration dependent (see comments below). The authors go on to include nucleosomes in one experiment to show that they prevent this mechanical stress from occurring by Sox2. Overall, this paper is of interest to the readers of nature communications. Specific comments are listed below.

Response: We thank the reviewer for the positive evaluation of our work. We have performed additional experiments and analyses that address the reviewer's comments, especially regarding the dependence of the condensation-driven mechanical tension on the Sox2 concentration.

1. Did the authors do varying concentration of sox2 in their assay to determine if this is a concentration dependent phenomena.

Response: This is an excellent suggestion. We varied the concentration of Sox2 (from 0 up to 500 nM, the highest that can be afforded in our experiments) and measured the force exerted on DNA via optical tweezers. We found that a higher Sox2 concentration results in a higher force readout (Fig. 3C). The cellular Sox2 concentration is estimated to be in the low micromolar range^{1,2}. Therefore, we speculate that the forces measured in our experiments are also relevant in the *in vivo* setting.

2. Do the authors know how many sox2 molecules are within these condensates?

Response: As described in the Methods section, we estimated the number of Sox2 molecules within each condensate by dividing the total fluorescence intensity of the Sox2 condensate by the average fluorescence intensity of Sox2 monomers (i.e. those non-specifically adsorbed to the surface in the same field of view).

3. Is the end breakage biologically relevant or just an example of the tension that Sox2 puts on the DNA? Is this concentration dependent, and if so, will this occur within a cellular setting?

Response: We view the DNA breakage in the TIRFM experiments as an example of the mechanical tension exerted by Sox2 condensates, which were quantified by the optical tweezers experiments. We found that the level of DNA breakage is indeed concentration-dependent (Supplementary Fig. 4C). As described in our responses to the other reviewers, such breakage likely resulted from a combination of Sox2-mediated condensation force and the destabilizing effects of the DNA intercalating dye used in the TIRFM experiments. Therefore, we do not think that it frequently occurs inside the cell. However, the mechanical effects of Sox2 condensation on DNA and chromatin, as well as the resulting force that we measured, are relevant in the physiological setting given the high cellular concentration of Sox2. We clarified this point in the revised manuscript.

4. The authors perform a control experiment with Cy3-H1 to determine if the condensation occurs with other DNA binding proteins. They perform this experiment at 150pM H1 and don't see as much condensation. However, with the sox2 they do these same experiments at 10nM. This raises concern that the results are arising from these experiments being performed at

substantially different concentrations. Did the authors test H1 at 10nM? The results in supplemental figure 5 indicate that H1 can also exert force on the DNA and it may be at a similar level as Sox2 if performed at the same concentration.

Response: We repeated the TIRFM experiment with 10 nM of Cy3-H1 and obtained similar results as those with 150 pM of H1. Thus, at the same concentration, H1 exerts much lower mechanical stress on DNA than Sox2. We updated the figure and Methods section with these new results.

5. Does sox2 dimerize with itself and therefore promote formation of these condensations through the IDRs?

Response: The elution profile of our Sox2 sample in gel filtration exhibited in a dominant peak consistent with a monomer. However, it has been reported that Sox2 can form a dimer in a DNA-dependent fashion, which requires the Group B homolog domain located at the C-terminus of Sox2 HMGB.³ We have shown in this paper that Sox2 without IDR (i.e. Sox2-HMGB) also forms co-condensates with DNA, albeit with slower kinetics and exerting attenuated mechanical stress compared to the full-length Sox2. Therefore, the dimerization activity of Sox2-HMGB may underlie its ability to form co-condensates with DNA, whereas the multivalent interaction mediated by Sox2's IDRs is likely responsible for its force generation effect. We added a discussion on these points in the revised manuscript.

6. The authors need to describe how they generated the histone samples for the single molecule experiments in more detail. They describe the expression and labeling, but not how they formed the final DNA/nucleosome substrate. They indicate that Nap1 is flowed in with the histone octamer to generate the DNA/nucleosome substrate. Did the authors verify if Nap1 is still present in the sample chamber when Sox2 was added?

Response: We apologize for this omission. After a 5-min incubation with immobilized DNA, Nap1 and unbound histone octamer were removed from the flow chamber by an excess wash of ~20x volumes of buffer before Sox2 was added. Based on the reduction in fluorescence background from the labeled histone octamer in solution, this procedure effectively removes any unbound protein from the flow chamber. This is now stated in the Methods section.

Reviewer #3 (Remarks to the Author):

In “Chromatin sequesters pioneer transcription factor Sox2 from exerting force on DNA”, Nguyen et al. present single-molecule studies of transcription factor Sox2 condensation on DNA. The authors use single-molecule fluorescence microscopy on tethered DNA to image Sox2 assembly into condensates on DNA, which reduces transverse fluctuations in DNA and destabilizes DNA to the point of breakage. This is presented as evidence for Sox2 condensates generating large forces. At steady-state, Sox2 condensates fuse and split—on the same DNA strand and between different DNA molecules—which confirms liquid-like behavior. Two Sox2 mutants, one without an intrinsically disordered region and one with a mutation in DNA-binding domain, are shown to slow phase-separation and reduce Sox2-induced DNA breakage. Using optical traps with

fluorescence microscopy, the authors measure a maximum force of 7 pN generated upon Sox2 condensation, an order of magnitude higher than that reported for other protein-DNA condensates. Sox2 condensates remain stably associated with the DNA even under mechanical tension high enough to unwind duplex DNA. Lastly, the authors present measurements of Sox2 on DNA pre-loaded with nucleosomes, showing that Sox2 colocalizes with nucleosomes, and their presence reduces Sox2-induced DNA breakage. These findings suggest that nucleosomes sequester Sox2 from exerting high forces on the genome, thereby resembling a mechanical “sink” that buffers stress within genomic DNA.

Overall the authors present an interesting finding which should be of broad interest to the scientific community, particularly researchers working on biomolecular condensates. The results are clearly and succinctly presented. The single-molecule fluorescence and optical tweezers experiments are first rate, and extract meaningful and novel physical properties of Sox2:DNA condensates. However, there are a number of important issues that need to be addressed. In particular, there are questions on the arguments for force generation and concerns over apparent inconsistencies between the fluorescence and optical tweezers measurements and their interpretation. The authors may be able to address these points and others (enumerated below) through additional measurements, including suggested controls. These questions need to be answered before a recommendation to publish can be made.

Response: We thank the reviewer for the comprehensive summary and overall enthusiasm on our work. We also appreciate the critical points raised by the reviewer. In our revised manuscript, we performed additional experiments and analyses to address these points.

Major points:

The authors should present control experiments with unlabeled, wild-type Sox2. In the TIRFM and trap assays, other readouts (e.g. YOPRO1 signal, force) can be used to detect DNA condensation, breakage, and/or force generation, i.e. independent of Sox2 labeling. One should expect these behaviors to be reproduced with unlabeled Sox2. This control is critical to determine if Cy5 labeling affects Sox2 activity and to rule out any photophysical effects.

Response: We thank the reviewer for suggesting this important control. We have performed additional experiments with unlabeled wild-type Sox2 and found that they recapitulate the same behaviors as found with labeled Sox2, including co-condensation with DNA, tension-induced DNA breakage in the TIRFM assay, and the magnitude of force generation measured by the optical tweezers assay. These results are summarized in the new Supplementary Fig. 4.

The authors argue that force generation by Sox2 condensates is responsible for DNA breakage, (observed by fluorescence imaging, Fig. 2). However, this mechanism does not seem plausible given the maximum condensation force of 7 pN (measured by optical trap, Fig. 3). It is unlikely that a force of 7 pN is sufficient to rupture duplex DNA. One possibility is that this force breaks the biotin-streptavidin linkages, but this is inconsistent with the observation of two fluorescence spots at the two tethering points after rupture (Fig. 2A). Another is that the DNA is highly nicked; in this case, 7 pN and thermal melting could perhaps lead to breakage, although this could

depend highly on the density of nicks. In either case, one would expect breakage to occur in the optical trap measurements; yet it is curious that the authors do not report this. Do they observe breakage in the trap? If not, this suggests that another mechanism is at play, or that the different conditions between TIRFM and trap measurements could have an effect (e.g. labeled vs. unlabeled DNA, surface bound DNA vs. levitated DNA). The authors should address this point at length and present a plausible mechanism for DNA rupture. Perhaps performing optical trapping experiments with the two mutants could help in establishing a mechanism.

Response: We thank the reviewer for bringing up this important issue, which was also raised by the other reviewers. We agree that the ~ 7 pN of force alone is unlikely to cause rupture of intact duplex DNA (as correctly pointed out by the reviewer, most breakage events occurred in the middle of the DNA- therefore they were not disruptions of the biotin-streptavidin linkage). We surmise that the DNA breakage observed in the TIRFM experiments resulted from a combination of the condensation force, the destabilizing effect of the DNA intercalating dye,^{4,5} and the DNA itself being nicked. This notion is based on a couple of new control experiments that we performed. First, we treated the λ DNA with T4 ligase to seal potential nicks and observed a much lower fraction of ruptured DNA after condensation (new Supplementary Fig. 5D). Second, we washed out the YOPRO1 dye after the initial imaging of DNA and before the addition of Sox2 and also found a much lower fraction of ruptured DNA (new Supplementary Fig. 5C).

We observed few breakage events in the optical tweezers experiments, which may be reasoned by 1) the DNA was not stained by YOPRO1 (we used force reading to detect the existence of the DNA tether; 2) the DNA was less photodamaged due to the different illumination geometry. Therefore, the force directly measured by the optical tweezers represents the mechanical tension exerted by the condensation alone without the other confounding factors in the TIRFM assay. These differences are now discussed at length in the revised manuscript.

Nevertheless, we still think that the fraction of DNA breakage observed in the TIRFM experiments is a useful proxy of the magnitude of mechanical tension for different proteins and substrates, given that they were conducted under the same conditions (laser power, dye concentration, etc.). This quantification complements the optical tweezers measurements that are by nature of low-throughput, which hinders us from testing all the constructs.

On a related point comparing the TIRFM and trap measurements, the authors should be able to estimate the tension on the DNA from its transverse fluctuations as measured in the TIRFM experiments (Fig. 1E-F). How do these numbers compare to the forces measurement by optical tweezers?

Response: In principle, one should be able to estimate the tension on the DNA based on the magnitude of its transverse fluctuations. However, this would require careful calibration, which is complicated by the heterogeneous end-to-end distance of the tethered DNA in the TIRFM assay, as well as the noise in the DNA fluorescence signal associated with the evanescent wave excitation. Moreover, because of the differences in the experimental conditions between the TIRFM and optical trap assays as detailed in our previous response, we refrained from making a direct comparison in terms of their force measurements. We believe that the forces directly

measured by optical tweezers are much more reliable, therefore we chose to report those values in the paper.

In the TIRFM measurements, the authors should present data on the effect of the surface on the condensate mobility. Is it possible that the decrease in transverse fluctuations in DNA is not due to increased tension, but due to the Sox2 condensates sticking to the surface and providing additional attachment points to the DNA? The authors could compare condensate mobility (e.g. diffusion constant) in the surface-based TIRFM measurements vs. solution-based trap measurements.

Response: We think that the decrease in transverse fluctuations in DNA is due to increased tension driven by the Sox2 condensates because: 1) the decrease in DNA fluctuations and the corresponding increase in the Sox2 condensate size are both gradual (Fig. 1C and D, Supplementary Fig. 3D, whereas they would be expected to be abrupt and immediate if they were due to surface sticking; 2) the timing of the decrease in DNA fluctuations is correlated with DNA breakage (Supplementary Fig. 3D) and DNA joining (Supplementary video 4), suggesting that it is due to force exertion by Sox2 condensates; 3) the mutant Sox2 constructs displayed much slower condensation and decrease in DNA fluctuations, whereas it is more difficult to rationalize why they would stick to the surface with different kinetics; and 4) the observation of Sox2 condensate fusion and splitting (Supplementary Fig. 3A, and Supplementary video 4) is inconsistent with surface sticking which should be irreversible. We have quantified the condensate mobility and found that it decreases over time, suggesting a maturation process (see our response to the next comment).

Moreover, as the reviewer pointed out, the solution-based optical tweezers experiments should rule out any potential surface effect, which is now mentioned in the revised manuscript. We observed correlated increases in tension and in Sox2 fluorescence signal (new Fig. 3B), supporting a direct connection between Sox2 condensation and force generation. We also observed Sox2 mobility in optical tweezers experiments that appeared to decrease over time (Fig. R1).

Figure R1. A representative kymograph showing Sox2 mobility on DNA in the optical tweezers assay.

Some protein-NA condensates have been reported to undergo a slow maturation where they transition to a more solid-like behavior. Are there any reports of this behavior for Sox2:DNA condensates? Do the authors see any evidence for maturation in their data (e.g. decreased condensate fusion events and overall mobility over time)? It may be that condensate maturation alters the strength of Sox2 binding to DNA, and perhaps this could result in breakage. The time frame for DNA breakage is in line with those of maturation in other systems.

Response: We thank the reviewer for bringing up this point. The maturation kinetics of Sox2-DNA condensates has not been reported before. We performed additional analyses of our data on Sox2 condensate mobility on DNA over time (new Supplementary Fig. 3B and C) and indeed found a decrease in mobility (quantified by the diffusion coefficient) on the timescale of 5-10 min. Notably, DNA breakage was predominantly observed when the condensates were in the liquid-like form (i.e. before 5 min), suggesting that force generation is attenuated once the condensates mature into more solid-like form.

Specific points:

The authors claim to measure the DNA content of the condensates (e.g. Fig. 1C), but how these values are determined is not well explained in Materials and Methods (see “Estimation of DNA content and Sox2 counts in a cluster”). The procedure outlined gives the number of Sox2 monomers, not the DNA content. Is the DNA content assumed to be equal to: (binding site size) x (# of monomers)? Is the DNA footprint of Sox2 well known? This should be clarified.

Response: We apologize for not making this clear. The DNA content was estimated based on the fluorescence intensity of the DNA staining dye within the condensates of a stretched λ DNA. This is now described in the Methods section. Even though the DNA footprint for specific Sox2 binding is relatively well-described,⁶⁻⁸ we chose not to infer the DNA content from the number of Sox2 monomers inside the condensate because Sox2 and DNA may form nonspecific, multivalent interactions therein (i.e. not at a 1:1 ratio).

The authors show that Sox2 binds DNA sequence-specifically (SI Fig. 2), but do not do much with this information. Does the position distribution of the condensates from fluorescence imaging (e.g. Fig. 1C or 3E) correlate with DNA sequence?

Response: This is an interesting question. We aligned the fluorescence intensity distribution of Sox2 condensates along the length of λ DNA from multiple tethers in the optical tweezers experiments. Our preliminary analysis shows that the positions of Sox2 condensates appear to be correlated with the positions of Sox2 binding motifs (Fig. R2). Due to the ambiguity associated with the left-right orientation of the λ DNA tether in our experiments, we refrained from making definitive conclusions in the current manuscript and plan to more rigorously examine this point in a follow-up study.

Figure R2. Position distribution of Sox2 intensity on bare λ DNA.

A) Average Sox2 intensity distribution along the length of λ DNA from optical tweezers experiments ($n_{\text{Sox2-50nM}}=4$, $n_{\text{Sox2-100nM}}=3$, $n_{\text{Sox2-150nM}}=3$, $n_{\text{Sox2-250nM}}=4$).

B) Histogram displaying the occurrence of the canonical Sox2 motif TTGT along the λ DNA genomic sequence.

C) Histogram displaying the occurrence of the extended Sox2 motif [C/A][A/T]TTGT.

On a related point, in Fig. 4E-F the authors show a correlation analysis for nucleosome and Sox2 condensate position. This is compared to a randomized control, but perhaps the data should be compared to Sox2 distribution on bare DNA, which is presumably not random based on the sequence specificity?

Response: We performed a preliminary analysis of Sox2 intensity distribution on nucleosomal DNA and found that nucleosomes dampen the dependency of Sox2 condensate positioning on DNA sequence (Fig. R3A). Rather, the Sox2 intensity is in general correlated with the nucleosome intensity (Fig. R3B). Without the ability to determine the orientation of DNA tethers at this time, we refrained from making definitive conclusions in our manuscript. We will rigorously examine the interplay between DNA sequence and nucleosome positioning in nucleating condensates in future studies.

Figure R3. Position distribution of Sox2 intensity on nucleosomal λ DNA.

A) Average Sox2 intensity distribution along tethered bare λ DNA (blue) or nucleosomal λ DNA (green) from optical tweezers experiments ($n_{\text{DNA}}=5$, $n_{\text{nuc}}=4$).

B) A representative nucleosomal λ DNA tether showing the alignment of Sox2 and nucleosome positions.

In addition:

1. For all figures, the authors should adjust the white scale bars such that they are all on a black background, not at the edge of the frames which makes them hard to see. Also, in the captions, they should describe what the arrows are indicating throughout (e.g. this is missing in Fig. 3D).

Response: Per reviewer's suggestion, we adjusted the scale bars and clarified what the arrows are indicating in the figure captions.

2. Consider adding the kymographs of SI Fig. 9 to Fig. 3.

Response: As suggested, we moved the kymographs of SI Fig. 9 to Fig. 3 (panels D-F).

3. In Fig. 3B, could the authors show the total fluorescence intensity measured at the same time? This would confirm that the tension increases as the condensates form. The authors could consider showing an average behavior instead of 1-4 representative traces (or show representative traces in grey and an average trace in color).

Response: We added the total fluorescence intensity (averaged from 4 representative traces) to the plot in Fig. 3B. The increase in fluorescence correlates with the increase in force. We thank the reviewer for this excellent suggestion.

References

1. Chen, J. *et al.* Single-molecule dynamics of enhanceosome assembly in embryonic stem cells. *Cell* **156**, 1274–1285 (2014).
2. Veneri, P. *et al.* Dynamical reorganization of the pluripotency transcription factors Oct4 and Sox2 during early differentiation of embryonic stem cells. *Sci. Rep.* **10**, 1–12 (2020).
3. Xia, P. *et al.* Sox2 functions as a sequence-specific DNA sensor in neutrophils to initiate innate immunity against microbial infection. *Nat. Immunol.* **16**, 366–378 (2015).
4. Biebricher, A. S. *et al.* The impact of DNA intercalators on DNA and DNA-processing enzymes elucidated through force-dependent binding kinetics. *Nat. Commun.* **6**, (2015).
5. Murade, C. U., Subramaniam, V., Otto, C. & Bennink, M. L. Interaction of oxazole yellow dyes with DNA studied with hybrid optical tweezers and fluorescence microscopy. *Biophys. J.* **97**, 835–843 (2009).
6. Soufi, A. *et al.* Pioneer transcription factors target partial DNA motifs on nucleosomes to initiate reprogramming. *Cell* **161**, 555–568 (2015).
7. Hou, L., Srivastava, Y. & Jauch, R. Molecular basis for the genome engagement by Sox proteins. *Semin. Cell Dev. Biol.* **63**, 2–12 (2017).
8. Schaefer, T. & Lengerke, C. SOX2 protein biochemistry in stemness, reprogramming, and cancer: the PI3K / AKT / SOX2 axis and beyond. *Oncogene* **39**, 278–292 (2020).

REVIEWERS' COMMENTS

Reviewer #1 (Remarks to the Author):

I appreciate the efforts of the authors. I think the authors did a good job of addressing the concerns I raised. The manuscript was further improved through the review process. I support publication of this paper in Nature Communications.

Reviewer #2 (Remarks to the Author):

The authors have addressed all my concerns. The manuscript is ready for publication.

Reviewer #3 (Remarks to the Author):

In their revision to “Chromatin sequesters pioneer transcription factor Sox2 from exerting force on DNA”, Nguyen et al. have done an excellent job of addressing the reviewer comments. The manuscript is recommended for publication in Nat. Comm.

One minor recommendation:

The authors carried out additional analyses showing a decrease in Sox2 condensate mobility on DNA over time (new Supplementary Fig. 3B and C). The authors should consider commenting and/or speculating on this in the discussion, especially in the context of condensate maturation. In their response to the reviews, the authors note that DNA breakage was predominantly observed when the condensates were in the liquid-like form, suggesting that force generation is attenuated once the condensates mature into more solid-like form. This point could be interesting to the reader.

Response to the reviewers' comments

Reviewer #1 (Remarks to the Author):

I appreciate the efforts of the authors. I think the authors did a good job of addressing the concerns I raised. The manuscript was further improved through the review process. I support publication of this paper in Nature Communications.

Reviewer #2 (Remarks to the Author):

The authors have addressed all my concerns. The manuscript is ready for publication.

Reviewer #3 (Remarks to the Author):

In their revision to “Chromatin sequesters pioneer transcription factor Sox2 from exerting force on DNA”, Nguyen et al. have done an excellent job of addressing the reviewer comments. The manuscript is recommended for publication in Nat. Comm.

One minor recommendation:

The authors carried out additional analyses showing a decrease in Sox2 condensate mobility on DNA over time (new Supplementary Fig. 3B and C). The authors should consider commenting and/or speculating on this in the discussion, especially in the context of condensate maturation. In their response to the reviews, the authors note that DNA breakage was predominantly observed when the condensates were in the liquid-like form, suggesting that force generation is attenuated once the condensates mature into more solid-like form. This point could be interesting to the reader.

Response: We thank the three reviewers for their constructive critiques that helped us improve this manuscript. Regarding the minor point from Reviewer #3, we have added a comment in the Discussion section: “We also observed that high force generation by Sox2 condensates, signified by the DNA breakage events, predominantly occurs early but diminishes as the condensates' mobility decreases over time. This observation indicates that the maturation of Sox2:DNA co-condensates from a liquid-like form to a solid-like one—akin to what was described in other systems³⁵—attenuates their force-generating capacity.”